# Optogenetics enables real-time spatiotemporal control over spiral wave dynamics in an excitable cardiac system

Rupamanjari Majumder[1†§], Iolanda Feola[1†], Alexander S Teplenin[1], Antoine AF de Vries[1], Alexander V Panfilov[2,3‡*], Daniel A Pijnappels[1‡*]

[1]Laboratory of Experimental Cardiology, Department of Cardiology, Heart Lung Center, Leiden University Medical Center, Leiden, The Netherlands; [2]Department of Physics and Astronomy, Gent University, Gent, Belgium; [3]Laboratory of Computational Biology and Medicine, Ural Federal University, Ekaterinburg, Russia

**\*For correspondence:**
Alexander.Panfilov@ugent.be (AVP);
D.A.Pijnappels@lumc.nl (DAP)

[†]These authors contributed equally to this work
[‡]These authors also contributed equally to this work

**Present address:** [§]Laboratory for Fluid Physics, Pattern Formation and Biocomplexity, Max Planck Institute for Dynamics and Self-Organization, Göttingen, Germany

**Competing interests:** The authors declare that no competing interests exist.

**Abstract** Propagation of non-linear waves is key to the functioning of diverse biological systems. Such waves can organize into spirals, rotating around a core, whose properties determine the overall wave dynamics. Theoretically, manipulation of a spiral wave core should lead to full spatiotemporal control over its dynamics. However, this theory lacks supportive evidence (even at a conceptual level), making it thus a long-standing hypothesis. Here, we propose a new phenomenological concept that involves artificially dragging spiral waves by their cores, to prove the aforementioned hypothesis in silico, with subsequent in vitro validation in optogenetically modified monolayers of rat atrial cardiomyocytes. We thereby connect previously established, but unrelated concepts of spiral wave attraction, anchoring and unpinning to demonstrate that core manipulation, through controlled displacement of heterogeneities in excitable media, allows forced movement of spiral waves along pre-defined trajectories. Consequently, we impose real-time spatiotemporal control over spiral wave dynamics in a biological system.
DOI: https://doi.org/10.7554/eLife.41076.001

## Introduction

Self-organization of macroscopic structures through atomic, molecular or cellular interactions is characteristic of many non-equilibrium systems. Such emergent dynamic ordering often reveals fundamental universalities (*Cross and Hohenberg, 1993*). One example is the occurrence of rotating spiral waves. Spiral waves are found in diverse natural systems: from active galaxies (*Schulman and Seiden, 1986*), to simple oscillatory chemical reactions (*Belousov, 1985*; *Zhabotinsky, 1991*), to social waves in colonies of giant honey bees (*Kastberger et al., 2008*), to Min protein gradients in bacterial cell division (*Caspi and Dekker, 2016*), and to the formation of vortices in fluids flowing past obstacles (*Karman, 1937*). While being beneficial to some systems, for example slime molds, where they guide morphogenesis, such activity has detrimental consequences for other systems including the heart, where they underlie lethal cardiac arrhythmias (*Davidenko et al., 1990*). Understanding the dynamics of spiral waves in order to establish functional control over a system, has intrigued researchers for many decades. It has been reported that irrespective of the nature of the excitable medium, spiral wave activity organizes around an unexcitable center (core), whose properties determine its overall dynamics (*Krinsky, 1978*; *Beaumont et al., 1998*). Theorists attribute such particle-like behavior of a spiral wave to an underlying topological charge, which controls its short-range interaction, annihilation, and the ability to form intricate bound states with other spirals (*Ermakova et al., 1989*; *Schebesch and Engel, 1999*; *Steinbock et al., 1992*).

**eLife digest** From a spinning galaxy to a swarm of honeybees, rotating spirals are widespread in nature. Even within the muscles of the heart, waves of electrical activity sometimes rotate spirally, leading to irregular heart rhythms or arrhythmia – a condition that can be fatal.

Irrespective of where they occur, spiral waves organize around a center or core with different biophysical properties compared to the rest of the medium. The properties of the core determine the overall dynamics of the spiral. This means that, theoretically, it should be possibly to completely control a spiral wave just by manipulating its core.

Now, Majumder, Feola et al. have tested this long-standing hypothesis using a combination of computer modeling and experiments with single layers of rat heart cells grown in a laboratory. First, the heart cells were genetically modified so that their electrical properties could be altered with light; in other words, the cells were put under optical control. Next, by using of a narrow beam of light, Majumder, Feola et al. precisely controlled the electrical properties of a small number of cells, which then attracted and supported a rotating spiral wave by acting as its new core. Moving the light beam allowed the core of the spiral wave to be shifted too, meaning the spiral wave could now be steered along any desired path in the cell layer.

Majumder, Feola et al. hope that these underlying principles may one day provide the basis of new treatments for irregular heartbeats that are more effective and less damaging to the heart than existing options. Yet first, more work is needed to translate these findings from single layers of cells to actual hearts.

DOI: https://doi.org/10.7554/eLife.41076.002

Rotational activity similar to spiral waves can also occur around small structural or functional heterogeneities (e.g. areas of conduction block). In this case, the dynamics of the rotating wave and its spatial position are determined by the location and properties of the heterogeneity. Thus, in theory, by controlling the position and size of spiral wave cores, one can precisely and directly control the dynamics of spiral waves in general. In order to achieve such control, it is therefore logical, to consider as a first step, possible core-targeting via the conversion of a free spiral wave to an anchored rotational activity. To this end, a detailed mechanistic study was performed by *Steinbock et al. (1993)*, who demonstrated the possibility to forcibly anchor meandering spiral waves in an excitable light-sensitive Belousov-Zhabotinsky (BZ) reaction system. Furthermore, *Ke et al. (2015)* demonstrated in a three-dimensional BZ reaction setting, that forced anchoring of scroll waves to thin glass rods, followed by subsequent movement of the rods themselves, could enable scroll wave relocation. On a broader perspective, this could have significant meaning for the heart, where controlling the dynamics of scroll waves could add to the treatment of cardiac arrhythmias sustained by such waves.

In cardiac tissue, the analogs of a classical spiral wave and a wave rotating around a heterogeneity, are, respectively, functional and anatomical reentry, both of which are recognized as drivers of arrhythmias. Interestingly, functional and anatomical reentrant waves are closely related to each other. Seminal findings by *Davidenko et al. (1991)* demonstrated that a drifting spiral wave could anchor to an obstacle and thereby make a transition from functional to anatomical reentry. Conversely, *Ripplinger et al. (2006)* showed that small electric shocks could unpin a reentrant wave rotating around an obstacle, bringing about the reverse transition from anatomical to functional reentry. *Nakouzi et al. (2016)* and *Zykov et al. (2010)* demonstrated that the transitions between anchored and free spiral states may be accompanied by hysteresis near the heterogeneities. Furthermore, *Defauw et al. (2014)* showed that small-sized anatomical heterogeneities could attract spiral waves from a close distance, and even lead to their termination if located near an unexcitable boundary. However, to date, all studies dedicated to spiral wave attraction and anchoring involved the presence of anatomically predefined, permanent heterogeneities, or continuous-in-time processes, thereby making it impossible to manipulate spiral wave cores in a flexible, systematic and dynamical manner. In the present study, we propose a new phenomenological concept to demonstrate real-time spatiotemporal control over spiral wave dynamics through discrete, systematic, manipulation of spiral wave cores in a spatially extended biological medium, that is cardiac tissue.

We establish such control through optogenetics (*Boyden et al., 2005*; *Bi et al., 2006*; *Deisseroth, 2015*; *McNamara et al., 2016*), which allows the creation of spatially and temporally predefined heterogeneities at superb resolution at any location within an excitable medium. Previous studies for example by *Arrenberg et al. (2010)*; *Bruegmann et al. (2010)*; *Jia et al. (2011)*; *Bingen et al. (2014)*; *Entcheva and Bub (2016)* and *Burton et al. (2015)*, demonstrate the power of optogenetics in cardiac systems. Thus, the same technology was chosen to strategically exploit fundamental dynamical properties of spiral waves, like attraction, anchoring and unpinning, to discretely and effectively steer spiral wave cores along any desired path within an excitable monolayer of cardiac cells. These findings are highly relevant for understanding non-linear wave dynamics and pattern formation in excitable biological media, as they enable, for the first time, real-time discrete dynamic control over processes that are associated with self-sustained spiraling phenomena, for example reentrant electrical activity, cAMP cycles and movement of cytosolic free $Ca^{2+}$, to name a few. In particular, in the heart, tight control of spiral waves may allow restoration of normal wave propagation.

## Results

Self-sustained spiral waves can be actively generated in most natural excitable media. In this study, we induced spiral waves of period $60 \pm 5$ ms in silico and of period $63 \pm 11$ ms in vitro in confluent monolayers of optogenetically modified neonatal rat atrial cardiomyocytes (see Materials and methods for details). Targeted application of light to the monolayers led to the sequential occurrence of two events: (i) creation of a spatially predefined temporal heterogeneity (i.e. a reversible conduction block) near the core of a spiral wave, and (ii) emergence of a wave from the spot of illumination. In a previous study (*Feola et al., 2017*), we demonstrated the possibility to create such a temporal heterogeneity with optical control over the size, location and duration of the block. Here, we show how creation of a light-induced block close to the core of a spiral wave, can attract the spiral wave tip, causing it to eventually anchor to the block. Subsequent movement of the light spot to different locations within the monolayer, results in dragging of the spiral wave along a predefined pathway of illumination, thereby giving rise to, what we call Attract-Anchor-Drag-based (AAD) control. *Figure 1A* illustrates a schematic diagram of the experimental setup that we used for

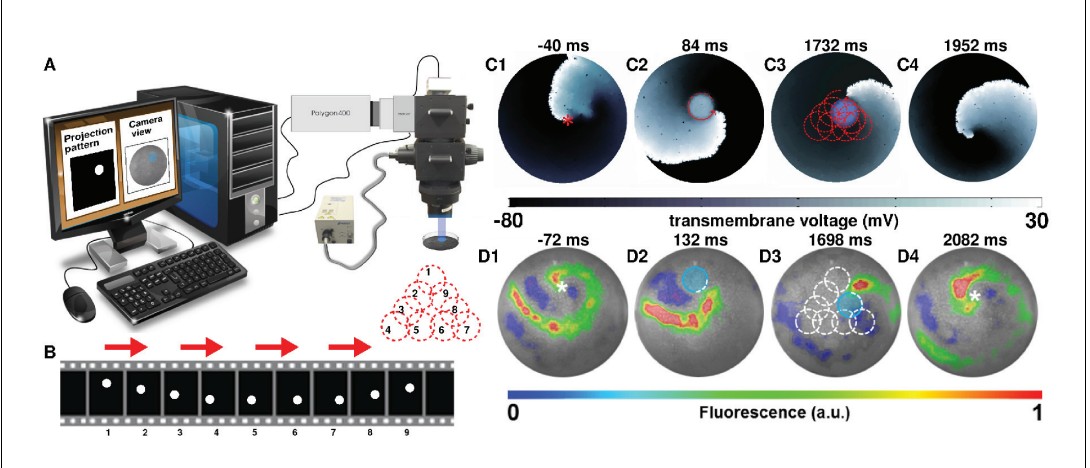

**Figure 1.** Attract-Anchor-Drag (AAD) control of a spiral wave core along a triangular trajectory. (**A**) Schematic diagram of our in vitro setup, showing how we project light patterns on an optogenetically modified monolayer. (**B**) The sequence of light spots that constitute the desired triangular trajectory of the spiral core. (**C1–C4**) In silico AAD control data recorded at subsequent times; (**D1–D4**) In vitro corroboration of our in silico findings (n = 9). In panels (**C**) and (**D**), the current location of the applied light spot is indicated with a filled blue circle, whereas, the schematic movement of the tip of the spiral wave, as it is anchored to the location of the light spot at previous time points, is indicated in each frame by means of dashed red (in silico) and white (in vitro) lines. The location of the phase singularity of the spiral wave is marked in the first and last frame of panels (**C**) and (**D**) with a red (in silico) or white (in vitro) asterisk. a.u., arbitrary units. A video demonstrating the process of dragging a spiral wave core along a triangular trajectory is presented in *Video 1*.
DOI: https://doi.org/10.7554/eLife.41076.004

our in vitro studies, whereas the sequence of light patterns used for real-time AAD control is described in *Figure 1B*. *Figure 1 (C1-4) and 1 (D1-4)* show representative examples of AAD control of spiral wave dynamics in silico and in vitro, respectively, where the spiral wave core is dragged along a predefined 9-step triangular path (see also *Video 1*). Our results demonstrate the possibility to capture a spiral wave, to manipulate its trajectory in a spatiotemporally precise manner (*Figures 1 C2–3 and D2*, and, finally, to release it in order to rotate freely again (*Figure 1 C4, D4*). Thus, we prove that it is indeed possible to directly and precisely manipulate the spatial location of a stable spiral wave core, and thereby overcome the constraints imposed by internal and external factors that shape the natural meander and drift trajectories. Such control is very important because it provides a direct and effective handle over the spatiotemporal patterning of the system, which would affect its general functionality. Natural systems currently suffer from the lack of such a handle (*Mikhailov and Showalter, 2006*; *Mikhailov and Loskutov, 1996*). Spiral wave dragging seeks to fill this lacuna at the simplest level.

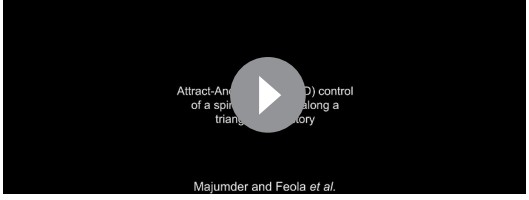

**Video 1.** Use of AAD control method to drag a spiral wave core along a triangular trajectory in silico and in vitro. **In vitro A** emphasizes the shape of the wave front. **In vitro B** shows the same data, but processed to show more clearly, what happens during the application of the light spots.
DOI: https://doi.org/10.7554/eLife.41076.003

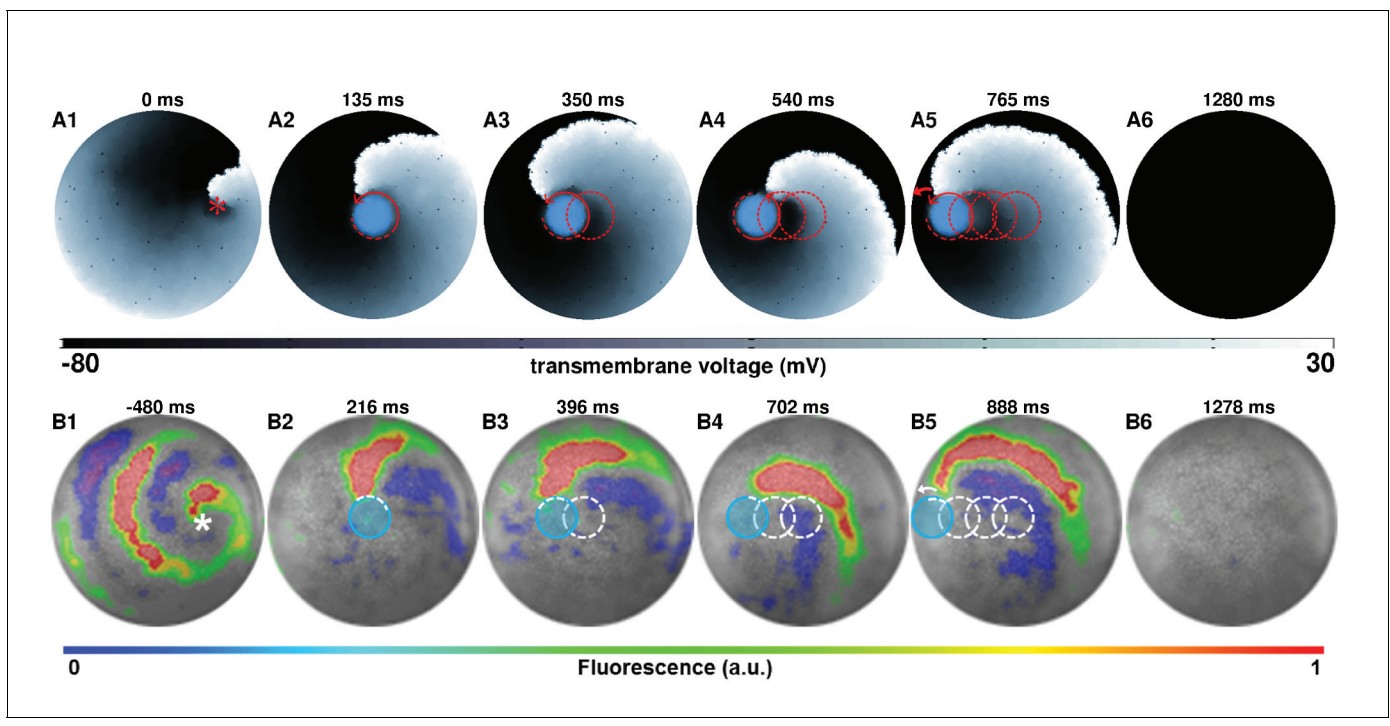

**Figure 2.** AAD control of a spiral wave core in favor of termination. The upper panel (**A1–A6**) shows successful removal of a spiral wave in silico, by capturing its core from the center of the simulation domain, and dragging it to the left boundary in a stepwise fashion ($\tau_{light} = 250$ ms and $\tau_{dark} = 1$ ms per spot, period of the reentry around the spot is $78 \pm 2$ ms). The lower panel (**B1–B6**) shows in vitro proof of the results presented in panel (**A**) (n = 9). For each light spot, the current location of the applied light spot is indicated with a filled blue circle ($\tau_{light} = 250$ ms and $\tau_{dark} = 15$ ms, period of the reentry around the spot is $90 \pm 12$ ms). The schematic movement of the tip of the spiral wave, as it is anchored to the location of the light spot at previous time points, is indicated in each frame by means of dashed red (in silico) and white (in vitro) lines. The location of the phase singularity of the spiral wave is marked in the first frame of each panel with a red (in silico) or white (in vitro) asterisk. a.u., arbitrary units. A video demonstrating the complete process of dragging a spiral wave core from the center of the monolayer to the left border, causing its termination, is presented in *Video 2*. (Time t = 0 ms denotes the moment when the light is applied).
DOI: https://doi.org/10.7554/eLife.41076.006

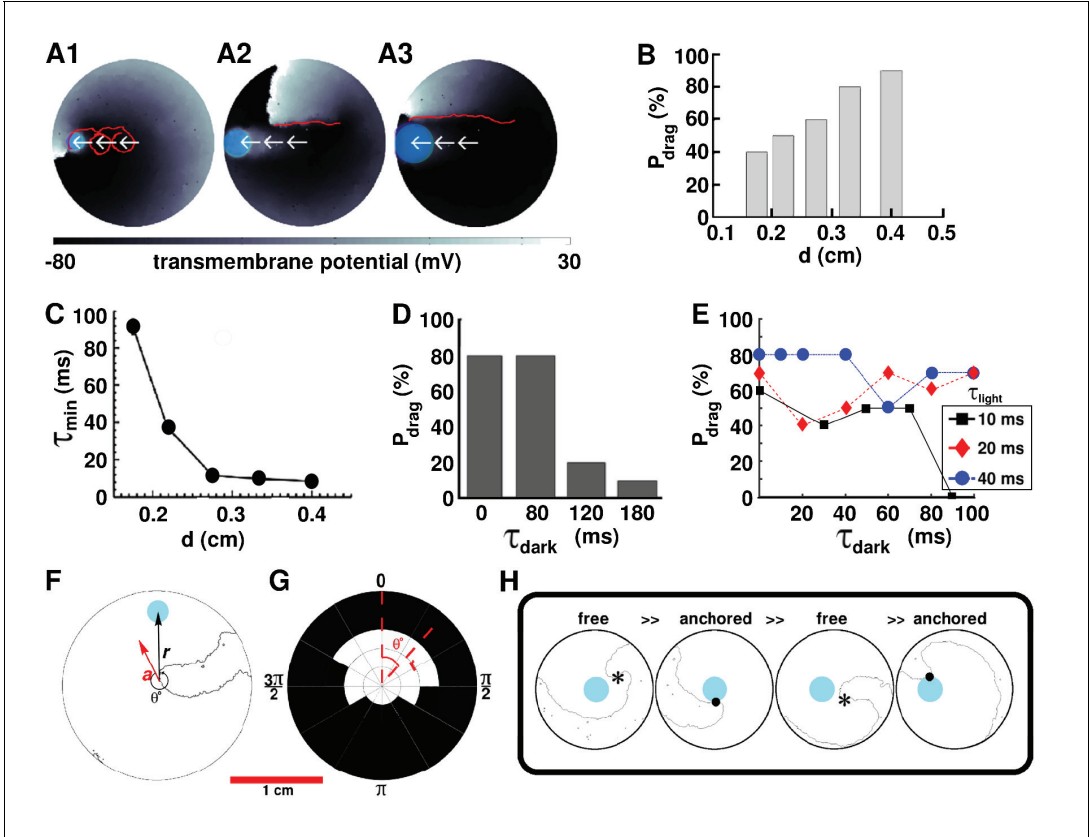

**Figure 3.** AAD control by continuous illumination of circular light spots of different sizes and durations. Panel A shows representative dragging events as a spiral wave core is relocated from the center of the simulation domain to the periphery, with light spots of diameter $d$=0.175, 0.275 and 0.4 cm, respectively. In each case, the real trajectories of the spiral tip is marked with solid red lines on top of the voltage map. When $d$ is large, the spiral tip exhibits a tendency to move along a linear path, as opposed to the cycloidal trajectory at small $d$. With $d = 0.275$cm, (B) probability of successful dragging ($P_{drag}$) increases, and (C) spiral core relocation time ($\tau_{min}$) decreases, with increasing $d$. In each of the afore-mentioned cases, the dragging process is illustrated with zero dark interval ($\tau_{dark}$) and minimal light interval ($\tau_{light}$). (D) With finite non-zero $\tau_{dark}$, $P_{drag}$ increases with shortening of $\tau_{dark}$, for light stimulation of afixed cycle length (200 ms). (E) Dependence of $P_{drag}$ on $\tau_{dark}$, for three different light intervals ($\tau_{light}$). (F) Schematic representation of the drag angle $\theta$ and the distance ($r$) between the starting location of the spiral tip and the location of the applied light spot. (G) Distribution of $P_{drag}$ at different $\theta$ and $r$, thresholded at 50%. (H) AAD control is effectuated by alternate transitions between functional (free wave) and anatomical (anchored wave) reentry. The phase singularity of the free spiral is marked with a black asterisk, whereas, the point of attachment of the anchored wave to the heterogeneity is marked with a black dot.

DOI: https://doi.org/10.7554/eLife.41076.007

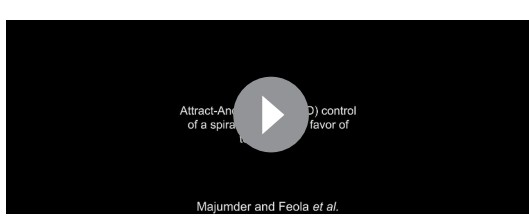

**Video 2.** Use of AAD control method to drag a spiral wave core to termination in silico and in vitro. **In vitro A** emphasizes the shape of the wave front. **In vitro B** shows the same data, but processed to show more clearly, what happens during the application of the light spots.

DOI: https://doi.org/10.7554/eLife.41076.005

An immediate application of such a process, that follows dynamic manipulation of spiral waves, involves using this technique to terminate complex spiral wave activity. We first focus on removing a single spiral wave in silico (*Figures 2 A1–5*; see also *Video 2*). With a five-step drag sequence of circular light spots of 0.275 cm diameter and 250 ms illumination time per spot, we demonstrate the possibility to remove a spiral wave from a monolayer, by capturing its core somewhere near the middle of the simulation domain and subsequently dragging it all the way to the unexcitable boundary on the left, causing the phase singularity to collide with the border and annihilate. The representative trajectory of

the spiral wave, as it is dragged to termination, is shown in *Figure 2 (A2-4)* with dashed red lines. The direction of movement of the spiral is indicated in each frame with red arrows. Inspired by the outcome of our in silico experiments, we tested the same principle in vitro, with a five-step drag sequence of circular light spots with similar characteristics as those applied in silico. Our in vitro findings corroborate the in silico results (*Figures 2 B1–5*; see also *Video 2*). The representative trajectory of this spiral wave, as it is dragged to termination, is shown in *Figure 2 (B2-4)* with dashed white lines. In each frame of *Figure 2*, the location of the light spot is marked with a transparent blue circle.

Having proven the possibility to drag single spiral waves to unexcitable tissue borders in favor of termination, we attempt to develop an in-depth in silico qualitative understanding of the parameters that play a crucial role in the dragging process. We focus on the previously described case: linear five-step dragging from the center of the simulation domain to a point on the periphery, with continuous illumination, and start by investigating the dependence of the probability of successful dragging ($P_{drag}$), on the diameter ($d$) of a continuously illuminated moving spot (i.e. without a finite time gap between successive applications of light). Our study reveals a positive correlation between $P_{drag}$ and $d$. At $d \leq 3$ mm, the minimum time ($\tau_{min}$) required for relocation of a spiral wave to the next position decreases abruptly with increasing $d$. However, at $d > 3$ mm, $\tau_{min}$ decreases slowly, trending towards a possible saturation value. These results are illustrated in *Figure 3*. Traces of the spiral tip trajectory (indicated by means of red lines on top of a representative frame depicted in *Figure 3 A1–A3*) demonstrate that the ease with which a spiral core can be dragged, from the center of the domain to a border, increases with increasing $d$. At small $d$, the spiral tip is dragged along a cycloidal path, in which, the diameter of the loop is $O(d)$. As $d$ increases, the spiral tends to translate linearly, without executing rotational movement about the core.

To investigate the conditions that allow spiral wave dragging with discrete light pulses, we apply a series of circular light spots of duration $\tau_{light}$. These spots appear sequentially in time, and are physically separated in space, with gradually decreasing distance from the boundary. For simplicity, we apply all spots along the same line, drawn from the center of the simulation domain to the boundary. We define the time gap between successive light pulses (when the light is off everywhere) as the dark interval $\tau_{dark}$. Thus, effectively, our optical stimulation protocol is periodic in time (with period $\tau_{light} + \tau_{dark}$), but not in space. We perform two sets of studies for the case of linear dragging. In the first set, we fix $\tau_{light} + \tau_{dark}$ at 200 ms and tune $\tau_{dark}$. In the second set, we fix $\tau_{light}$ (three different values) and tune $\tau_{dark}$. Our results indeed demonstrate the occurrence of AAD control with discrete illumination ($\tau_{dark} \neq 0$ ms). *Figure 3D* shows, for example, that at $\tau_{dark} = 0$, 80, 120, and 180 ms, respectively, $P_{drag}$ decreases from 80% to 20% to 10%. Interestingly, $\tau_{dark} = 80$ ms appears to be as effective as continuous illumination ($\tau_{dark} = 0$ ms, *Figure 3D*). Thus, our findings confirm that, for a fixed cycle length of optical stimulation by a spot of light of diameter $d = 0.275$ cm, $P_{drag}$ increases with decreasing $\tau_{dark}$.

If, however, we tune to explore the dependence of $P_{drag}$ on $\tau_{light}$ at flexible cycle lengths, there is no clear relationship between $P_{drag}$ and $\tau_{light}$ (*Figure 3E*), despite an increase in the ease of attraction of the spiral tip towards the illuminated spot at short $\tau_{dark}$. In general, $\tau_{light} \simeq O(0.1\tau_{dark})$ does not lead to effective AAD control, whereas, $\tau_{light} \simeq \mathcal{O}(10\tau_{dark}, \tau_{dark} \neq 0)$ does. However, exceptional cases also exist. For example, (i) $P_{drag}$ can be large (i.e. 70%) when a short light pulse ($\tau_{light} = 20$ ms) is followed by a long dark interval ($\tau_{dark} = 100$ ms); and, (ii) combinations of abbreviated $\tau_{light}$ and $\tau_{dark}$ (i.e. $\tau_{light} = 20$ ms, $\tau_{dark} = 20$ ms) can lead to lower $P_{drag}$ than pulse combinations with shorter or longer $\tau_{dark}$.

We find that probability of attraction also depends on the phase of the spiral wave rotation at the moment of light exposure. This dependence, in simple terms, can be quantified as dependence on the angle between the vector along the instantaneous direction of motion of the spiral tip $\vec{a}$ and the vector $\vec{r}$ of the displacement of the center of the circular light spot, in polar coordinate system. This is illustrated in *Figure 3F*. Our results demonstrate that spiral wave dragging occurs most efficiently if the spot of light is applied sufficiently close to the location of the spiral tip and within an angular spread of $\Delta\theta_X$ in the direction of drift of the spiral core (*Figure 3G*). Here, $X$ denotes the cutoff probability for the occurrence of dragging, that is $\Delta\theta_{50\%}$ refers to an angular spread of $\Delta\theta$ for which $P_{drag} \geq 50\%$. A spiral wave cannot be dragged when $r > 5.63$ mm. When 4.7 mm $< r <$ 5.63 mm, $\Delta\theta_{50\%} = 150°$. $P_{drag}$ increases to 240° when 3.44 mm $< r <$ 4.7 mm, and to 360° when $r < 3.44$ mm.

Furthermore, to develop a detailed understanding of the physical mechanisms involved in spiral wave dragging, the nature of the process itself is analyzed. Our in silico findings indicate that successful control necessitates the spiral wave to make alternate transitions between functional (free) and anatomical (anchored) types of reentry. When a spot of light is applied reasonably close to the core of a spiral wave, within the allowed $\Delta\theta$, it creates a region of light-induced depolarization that attracts and anchors the core of the free spiral, thereby effectuating a transition from functional to anatomical reentry. When the light spot is now moved to a different location, still within the basin of attraction of the first light spot, the previously depolarized region recovers, forcing the anchored spiral to unpin and make a reverse transition to functional reentry. The larger the value of $\tau_{dark}$, the more visible the transition. Finally, at the new location of the light spot, a zone of depolarization is created, which either attracts and anchors the spiral tip, or produces a new wave of excitation that replaces the existing spiral wave with a new one, still followed by attraction and anchoring. A sequence of in silico voltage maps in support of the proposed mechanism, is shown in *Figure 3H*.

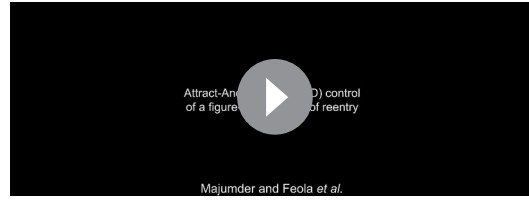

**Video 3.** Use of AAD control method to terminate figure-of-eight type of reentry in silico and in vitro. **In vitro A** emphasizes the shape of the wave front. **In vitro B** shows the same data, but processed to show more clearly, what happens during the application of the light spot.
DOI: https://doi.org/10.7554/eLife.41076.008

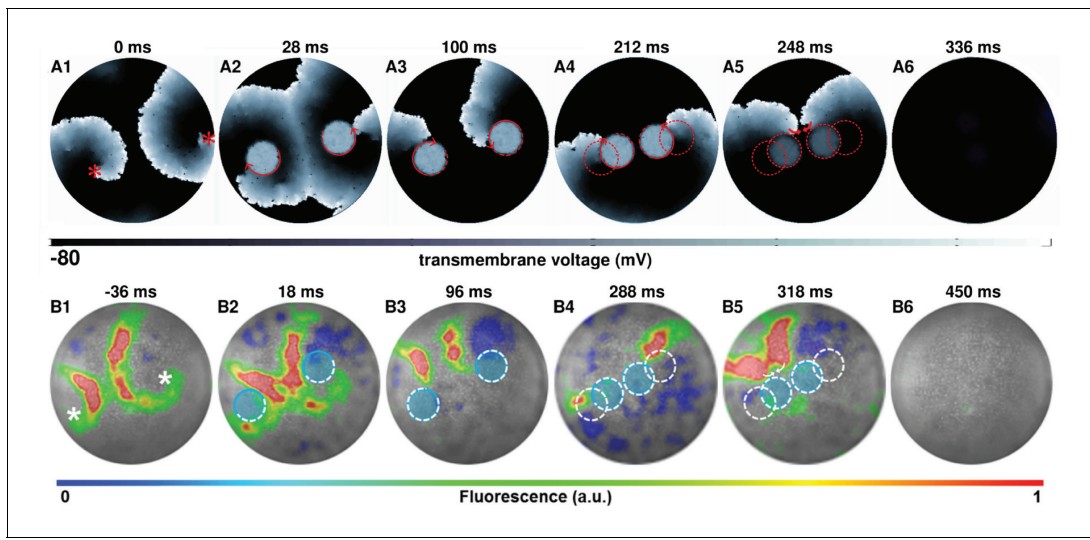

**Figure 4.** AAD control of a pair of spiral wave cores in favor of termination of figure-of-eight type reentry. The upper panel (**A1–A6**) shows representative in silico voltage maps of the drag-to-termination process, at subsequent times. The lower panel (**B1-B6**) provides experimental (in vitro) validation of the findings presented in panel (**A**) (n = 6). In each case, the locations of the phase singularities are marked on the first frame of each panel with a red (in silico) or a white (in vitro) asterisk. The current location of the applied spot of light is indicated with a filled blue circle. The schematic movement of the tip of the spiral wave, as it is anchored to the location of the light spot at previous time points, is indicated in each frame by means of dashed red (in silico) and white (in vitro) lines. The direction of dragging is indicated in panel (**A**) with red arrows. In our studies, we terminated figure-of-eight type reentry by dragging 2 spiral cores toward each other to make them collide and annihilate. a.u., arbitrary units. A video demonstrating AAD control (with eventual termination) of a figure-of-eight type reentry is presented in *Video 3*. (Time t = 0 ms denotes the moment when the light is applied.).
DOI: https://doi.org/10.7554/eLife.41076.009

The following figure supplement is available for figure 4:

**Figure supplement 1.** AAD control of spiral wave cores to accomplish termination of a complex figure-of-eight type reentry, where the constituting spirals rotate slightly out of phase with each other.
DOI: https://doi.org/10.7554/eLife.41076.010

With a detailed understanding of the parameters involved in the process of spiral wave dragging, we explore the possibility to apply this phenomenon to tackle more challenging problems, such as, termination of complex patterns of reentry. We specifically target two patterns: (i) figure-of-eight type reentry, and (ii) reentry characterized by multiple spiral waves, that is multiple phase singularities. In silico, we find that figure-of-eight type reentry can be removed efficiently by dragging the cores of both spiral waves towards each other, till their phase singularities collide, leading to self-annihilation.

*Figure 4 (A1-6)* (see also *Video 3*) show subsequent steps in the process of removal of a figure-of-eight type reentry via AAD control, when the reentry pattern comprises two spirals of opposite chirality, rotating in phase with each other. *Figure 4 (B1-6)* (see also *Video 3*) show successful experimental (in vitro) validation of our in silico findings, following the same termination protocol. When the figure-of-eight type reentry comprises two spirals of opposite chirality that do not rotate exactly in phase, the strategy should be to capture the cores of the pair of spirals with light spots of unequal sizes, so as to compensate for the difference in phases, at the very first step of the control method. A representative example of such a scenario is presented in the *Figure 4—figure supplement 1*.

In order to terminate reentry characterized by multiple phase singularities, we try different approaches. *Figure 5 (A1-7) and (B1-7)* (see also *Video 4*) illustrate two examples of such attempts in silico with reentry characterized by three and seven phase singularities, respectively. With three spiral cores, we follow a strategy which is a combination of the cases presented in *Figures 4* and *2* that is we capture all three cores (*Figure 5A2*), drag them toward each other (*Figure 5A3*), reduce complexity via annihilation of two colliding phase singularities (*Figure 5 A4–5*), and drag the remaining spiral core to termination via collision of its phase singularity with the unexcitable boundary (*Figures 5 A6–7*). However, with higher complexity, there is no unique optimal approach that leads to

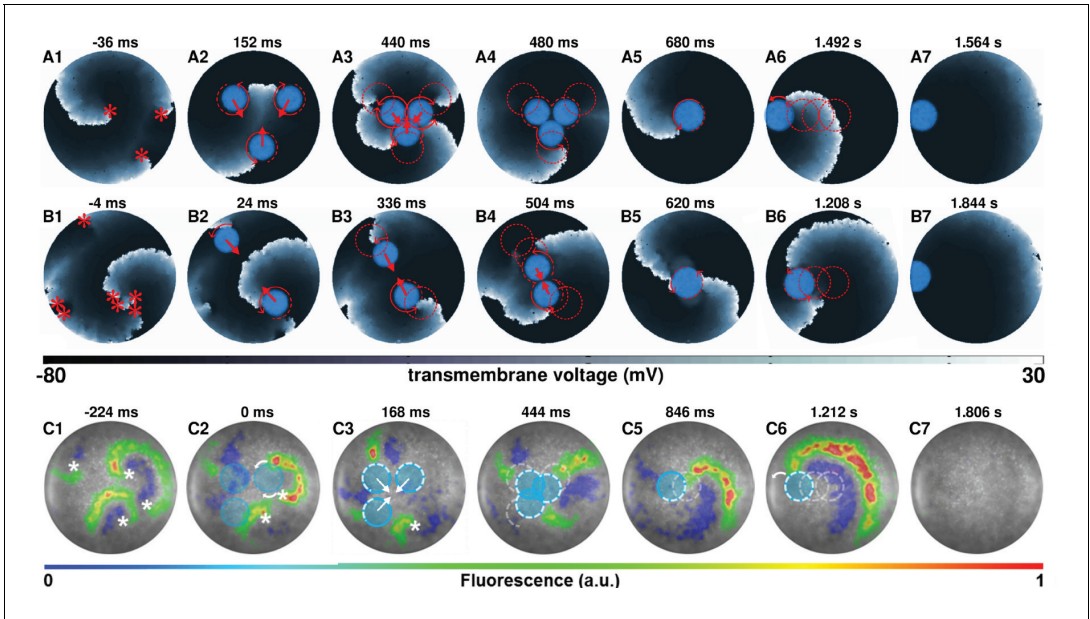

**Figure 5.** AAD control of multiple spiral wave cores in favor of termination of complex reentrant patterns. Panels (**A**) and (**B**) show representative in silico voltage maps of the drag-to-termination process for stable reentrant activity with three and seven phase singularities, respectively. Panel (**C**) shows representative in vitro voltage maps of the drag-to-termination process for stable reentrant activity with four phase singularities. Number and the position of the light spots, indicated with blue filled circles, are strategically applied to first capture the existing spiral waves, and then drag them to termination, following the same principle as in *Figure 2 (A5-6) and (B5-6)*. In the process, it is possible to reduce the complexity of the reentrant pattern through annihilation of some of the existing phase singularities by new waves emerging from the location of the applied light spot. The schematic movement of the tip of the spiral wave, as it is anchored to the location of the light spot at previous time points, is indicated by means of dashed red (white) lines in silico (in vitro), and the direction of dragging is illustrated with red (white) arrows. *Video 4* shows a video of computer simulations demonstrating AAD control of complex reentrant patterns with three and four spiral waves, in favor of their eventual termination. *Video 5* shows the same process, in vitro, but for stable reentry with four-phase singularities. (Time t = 0 ms denotes the moment when the light is applied).
DOI: https://doi.org/10.7554/eLife.41076.013

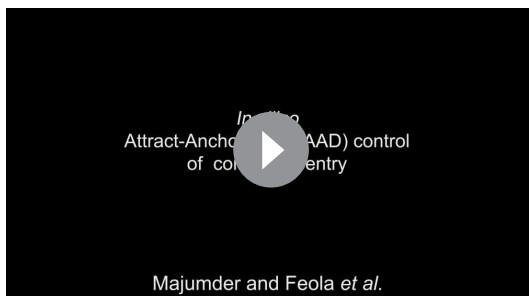

**Video 4.** Use of AAD control method to terminate complex reentry in silico, with 3 (left) and 4 (right) spiral waves.
DOI: https://doi.org/10.7554/eLife.41076.011

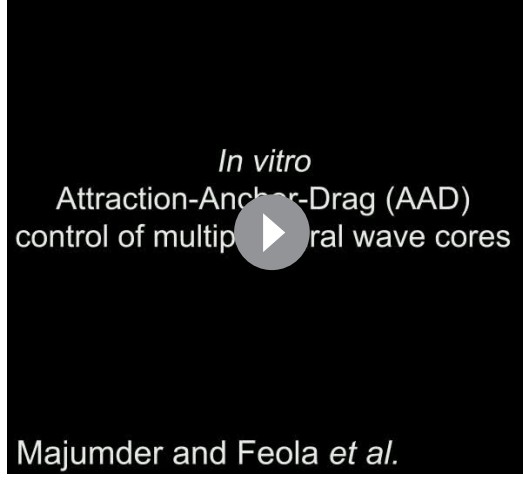

**Video 5.** Use of AAD control method to terminate complex reentry (four spiral waves) in vitro. This video was produced in the same manner as described above for **In vitro A**.
DOI: https://doi.org/10.7554/eLife.41076.012

the termination of reentry. In the example presented in *Figure 5 (B1-7)*, we rely on reducing the complexity of the reentrant pattern itself, via strategic application of two light spots, prior to the actual dragging process. New waves emerging from the locations of the applied light spots interact with the existing electrical activity in the monolayer to produce new phase singularities, which collide with some of the preexisting ones to lower the complexity of the reentrant pattern. With reduced complexity, we drag the spiral cores toward each other, till they merge to form a single two-armed spiral, anchored to the applied light spot. We then drag this spiral to the boundary of the monolayer causing its termination. A similar strategy applied in vitro enabled successful termination of a complex reentry pattern characterized by four phase singularities. This is illustrated in *Figure 5 (C1-7)*. At the instant when the first set of light spots is applied, reentry is characterized by four phase singularities (*Figure 5C1*). Note that, of the four phase singularities, two coincide with the locations of two of the applied light spots and two are located away from the third light spot (*Figure 5C2*). These first two phase singularities anchor immediately to the applied light spots. However, the other two interact with the new wave generated by the third applied light spot and reduce to one phase singularity, which remains free (unanchored) (*Figure 5C3*). When the next set of light spots are applied (in a complex pattern of three overlapping circles), this phase singularity displays attraction to the new light spot, thus effectively producing a three-armed spiral (*Figure 5C4*). Finally, when the complex light pattern is replaced by a smaller circular spot of light, the three-armed spiral loses two of its arms through interaction with the emergent new wave from the location of the applied light spot (*Figure 5C5*). The single spiral can then be dragged to termination (*Figures 5 C6–7*) as in *Figure 2*.

Taken together, our results show that core positions of spiral waves can be strategically manipulated with the help of optogenetics to establish direct spatiotemporal control over a highly nonlinear system. We demonstrate that application of well-timed and spaced light pulses can result in successful dragging of a spiral wave from one location to another within a monolayer, along any desired path and even to cause its termination. Spiral wave dragging can also be employed to terminate complex reentrant patterns characterized by multiple phase singularities, through appropriate AAD control. However, the probability of its occurrence is subject to two key parameters: (i) size $d$ and (ii) drag angle ($\Delta\theta$).

## Discussion

Spiral waves occur in diverse natural excitable systems, where they impact the functioning of the system in a beneficial or detrimental manner. Given their dynamic and unpredictable nature, direct control over these nonlinear waves remains a longstanding scientific challenge. While it is an established theory that the properties of the core of a spiral wave determine its overall dynamics (*Krinsky, 1978*;

*Beaumont et al., 1998*), the corollary that manipulation of spiral wave cores should lead to full spatiotemporal control over wave dynamics in excitable media, appears to be unexpectedly non-obvious and understudied (see below). In this paper, we prove the aforementioned consequence using the remarkable features of optogenetics. Our study is one of the first to demonstrate the establishment of real-time spatiotemporal control over spiral wave dynamics in spatially extended biological (cardiac) tissue. The only other study that shows effective dynamic spatiotemporal control (as opposed to elimination) of spiral waves is that of (*Burton et al., 2015*). However, *Burton et al. (2015)* use dynamic control to modulate spiral wave chirality, which is markedly different from what we study and prove, namely spiral wave dragging.

In this research, we demonstrate a new phenomenological concept to steer spiral waves in excitable media, thereby enabling precise and direct spatiotemporal control over spiral wave dynamics. This so-called AAD control involves a feedback interaction between three fundamental dynamical properties of spiral waves: attraction, anchoring and unpinning. In the past, researchers also employed different kinds of spatiotemporal feedback interactions to control spiral wave dynamics, albeit in non-biological media (*Sakurai et al., 2002*). Their methods, while effective, relied on delivering timed pulses falling in defined phases of the spiral. Although feasible in simple chemical systems, phase-based pulse delivery can be quite challenging in complex biological tissue, where temporal precision is a major subtlety and therefore difficult to achieve. The study by *Ke et al. (2015)* bypasses this issue by forcing scroll wave filaments to anchor to physical heterogeneities, which can be moved subsequently to steer the waves. However, *Ke et al. (2015)* demonstrate a process that is continuous in time, as opposed to the discrete nature of our method, which makes it more flexible.

Previous studies have explored the concept of spiral attraction in human cardiac tissue (*Defauw et al., 2014*). According to these findings, the mechanism of attraction of spiral waves to localized heterogeneities is a generic phenomenon, observed among spiral waves in heterogeneous tissue. Some studies (*Defauw et al., 2014*; *Panfilov and Vasiev, 1991*; *Rudenko and Panfilov, 1983*) show that spiral waves tend to drift to regions with longer rotational periods, that is where the duration of an action potential is longer than elsewhere in the medium. This is in consonance with our simulations, where application of a spot of light to a monolayer of excitable, optogenetically modified cardiomyocytes causes the cells in this region to depolarize, bearing a direct influence on the action potential duration of cells in the immediate neighborhood. Once attracted, the spiral anchors to the location of the light spot, thereby making a transition from functional to anatomical reentry. Again, when the light at the current location is switched off and a new location is illuminated, in the next step of the drag sequence, the spiral reverts back to functional reentry, from its anatomical variant. If the new location is such that the spiral core overlaps with the basin of attraction of the optogenetically depolarized area, then it anchors to the new location and makes a transition back to anatomical reentry. Thus, AAD control is associated with repeated alternative transitions between functional and anatomical reentry. However, if the core of the spiral wave does not overlap with the basin of attraction of the next location in the drag sequence, or, if the drag sequence is executed too fast for the spiral to establish a stable core at subsequent new locations, then the spiral detaches, breaks up, or destabilizes without being successfully dragged.

A mechanism, somewhat along the same lines as AAD control, was demonstrated by *Krinsky et al. (1995)*, using topological considerations. When an existing vortex interacts with a circular wave emerging from a stimulus applied close to the vortex core, they fail to annihilate each other. However, the emergent wave quenches the vortex, resulting in its displacement by the order of half a wavelength. This displacement may occur in any direction, as demonstrated by Krinsky et al. In our system, we observe similar emergence of a new (circular) wave from the temporal heterogeneity. This new wave fails to annihilate the existing spiral wave, in line with Krinsky's findings, but leads to the displacement of the core toward the temporal heterogeneity, indicating the existence of a factor of attraction, which determines the direction of drift.

In another study (*Guo et al., 2010*), control of spiral wave turbulence was investigated in a self-regulation system that spontaneously eliminated vortex cores, by trapping the latter with localized inhomogeneities. This study used computer simulations and a light-sensitive BZ reaction with a Doppler instability. Movement of a spiral wave along a line of predefined obstacles was also observed by *Andreev (2005)*. However, unlike Krinsky et al. and *Guo et al. (2010)*, their study uses inactive defects, which do not lead to quenching of the spiral wave. Further topological

considerations with respect to movement of spiral waves can be found in the seminal work of *Winfree and Strogatz (1984)*. In our studies of spiral wave dragging with light spots of different sizes (*Figure 3*), a small-sized spot leads to dragging via a cycloidal trajectory, whereas, a large spot is associated with a linear trajectory. In our subsequent investigations, however, cycloidal trajectories were observed even for large spots. This is because, in the data presented in *Figure 3*, we investigate dragging at the fastest possible rate. The drag trajectories obtained in that study apply for minimal relocation time ($\tau_{light} = \tau_{min}$, $\tau_{dark} = 0$ ms). The larger the value of $d$, the bigger is the area of light-induced depolarization. Consequently, the higher is the chance for the tip of the spiral wave to fall within the basin of attraction of the illuminated area, causing it to remain anchored to the spot at all times. When the spot of light is moved with $\tau_{dark} = 0$ ms and $\tau_{light} = \tau_{min}$, the location of the new light spot forms a composite anchor point in combination with the location of the previous light spot; the effective depolarized region resembles an elongated ellipse, with elongation along the direction of movement of the light spot. This causes the spiral tip to trace a nearly linear path, along the boundary of the elongated ellipse. However, at slower dragging rates, that is longer $\tau_{light}$, as used in the subsequent studies (*Figure 3*), the spiral wave gets ample time to execute one or more complete rotations around the region of light-induced depolarization, before being relocated to the next spot. Hence, the cycloidal trajectory at large $d$. The tendency for the spiral tip to move along a cycloidal trajectory at small $d$ can be explained by the establishment of a very small basin of attraction, which results in an increased demand for $\tau_{min}$. Thus the spiral requires execution of at least one complete rotation around the region of light-induced depolarization before it can relocate to the next spot.

These findings are in consonance with the fundamental view of excitable media as dynamical systems. Based on their experimental, computational and theoretical studies, *Zykov et al. (2010)* and *Nakouzi et al. (2016)* reported the involvement of bistability and hysteresis in the transitions from anchoring to unpinning of a wave rotating around an anatomical hole. Bistability, or coexistence of two dynamical attractors in the system, indicates dependence of the outcome of the transition on initial conditions. In our case, the most important choice of initial conditions can be restricted to the phase of the freely rotating spiral wave or the rigidly anchored reentrant wave. Thus, proper timing of the applied perturbation is crucial regarding the efficacy of control. In this sense our results deceptively remind us of the phenomenon of spiral wave drift, which can be induced by periodic modulation of excitability (*Steinbock et al., 1993*). However, such drift arises from an interplay between Hopf bifurcation of spiral wave solutions, and rotational and translationally symmetric planar (Lie) groups (*Wulff, 1996*). In our studies, we rely on completely different nonlinear dynamical phenomena, namely, bistability of the rotating wave solutions and its interplay with spatial and temporal degrees of freedom of the dynamically applied light spot (i.e. its position, size, period and timing). This combination gives rise to novel interesting spiral wave dynamics, which opens broad avenues for further theoretical and experimental investigation.

Although apparently straightforward, and, to some extent, predictable on the basis of generic spiral wave theory, our results are not at all obvious. Compared to other systems, such as the BZ reaction, cardiac tissue (i) possesses an inherently irregular discrete cellular structure, underlying the absence of spatial symmetries, (ii) lacks inhibitor diffusion and (iii) has different dimensions. Moreover, since the probability of attraction of a spiral wave to a heterogeneity is determined by the initial conditions and by the strength of successive perturbations, it is not a foregone conclusion that discrete application of light spots outside the core of a spiral wave should always end up in an anchored state. By showing, in cultured cardiac tissue, that spiral wave dynamics can be controlled by targeting the spiral wave core, we have been able to experimentally confirm pre-existing ideas about the behavior of spiral waves in complex systems.

Previous studies also demonstrated alternative methods to 'control' spiral-wave dynamics in excitable media, for example, by periodic forcing to induce resonant drift (*Agladze et al., 1987*; *Biktasheva et al., 1999*; *Agladze et al., 2007*), or by periodic modulation of excitability to manipulate spiral tip trajectories (*Steinbock et al., 1993*). These studies do demonstrate the potential to drive a spiral wave out of an excitable medium, or to force spiral waves to execute complex meander patterns, thereby making landmark contributions to the knowledge of pattern formation or even to cardiac arrhythmia management. However, the principal limitation of these methods is that they are indirect, giving reasonable control over the initial and final states of the system, with little or no

control in between. It is herein where lies the advantage of AAD control, which allows precise and complete spatiotemporal control over spiral wave dynamics in two-dimensional excitable media.

## Clinical translation

Since we demonstrate AAD control method in a cardiac tissue system, a logical question would be, how to envision the application of this principle to the real heart in order to treat arrhythmias? Currently this topic faces major challenges. The practical application of optogenetics in cardiology is, in itself, a debatable issue. However, with recent advances in cardiac optogenetics (*Nussinovitch and Gepstein, 2015*; *Crocini et al., 2016*; *Nyns et al., 2017*; *Bruegmann et al., 2018*; *Boyle et al., 2018*), the future holds much promise. Firstly, we envision the usage of AAD control method in treating arrhythmias that are associated with scroll waves. Since the penetration depth of light in cardiac tissue is relatively short (*Bruegmann et al., 2016*), full transmural illumination might be challenging, particularly in ventricles of large mammals like pig, monkey or human. There, AAD control may provide a powerful tool to regulate scroll wave dynamics by epicardial or endocardial illumination. Furthermore, we expect the method to prove most useful when dealing with 'hidden' spiral waves, that is spiral waves in remote locations of the heart that are unaccessible by ablation catheters. Ideally, one should build upon the concept introduced by *Entcheva and Bub (2016)*. With live spacetime optogenetic actuation of the electrical activity in different parts of the heart, the first step is to detect the location of the instability. Next, one can use a catheter with an in-built LED to attract the scroll wave filament and steer it towards the nearest tissue border for termination. The advantage of this method lies in that one does not require to ablate, and thereby destroy, excitable cardiac tissue, thus avoiding the possibility to create permanent damage to the heart. In addition, as our study demonstrates, anchoring of the spiral core (scroll filament) can occur even if the 'precise' location of the core is not identified. Lastly, the discrete nature of our method allows temporal flexibility in steering the spiral core (scroll filament), in that, temporary loss of communication between the catheter and the spiral core will not lead to failure of the technique in general. The reversible nature of the AAD control technique makes it unsuitable for terminating arrhythmias that rely on the establishment of a permanent conduction block as can be produced via conventional catheter ablation. However, this special feature of AAD may come with certain unique advantages that should be explored in more detail in future studies. For example, in younger patients that are expected to undergo periodic repetitive ablation for termination of reoccurring arrhythmias of unknown origin, the non-destructive nature of AAD may prove to be more desirable than the cumulative widespread destruction of cardiac tissue by radiofrequency or cryoballoon ablation. Alternatively, other methods could be developed and explored for AAD control without the need of optogenetic modification, while still relying on the creation of spatiotemporally controlled heterogeneities for attraction, anchoring and dragging of spiral waves.

In this study, we focus on spiral waves in cardiac excitable media, as these abnormal waves have been associated with lethal heart rhythm disturbances, while their management and termination remain a serious challenge. The insights gained from our results, as well as the AAD control method itself, may not only improve our understanding of spiral wave's dynamics in favor of restoring normal cardiac rhythm, but also create incentive to explore these principles in other excitable media prone to spiral wave development.

## Materials and methods

The electrophysiological properties of neonatal rat atrial cardiomyocytes were modeled according to *Majumder et al. (2016)*, whereby, the transmembrane potential $V$ of a single cell evolved in time as follows:

$$\frac{dV}{dt} = -\frac{I_{ion} + I_{stim}}{C_m} \tag{1}$$

$I_{ion}$ is the total ionic current, expressed as a sum of 10 major ionic currents, namely, the fast $Na^+$ current ($I_{Na}$), the $L-type$ $Ca^{2+}$ current ($I_{CaL}$), the inward rectifier $K^+$ current ($I_{K1}$), the transient outward $K^+$ current ($I_{to}$), the sustained outward $K^+$ current ($I_{Ksus}$), the background $K^+$ ($I_{Kb}$), the background $Na^+$ ($I_{Nab}$) and the background $Ca^{2+}$ ($I_{Cab}$) currents, the hyperpolarization-activated funny

current ($I_f$), and the acetylcholine-mediated $K^+$ current ($I_{K,ACh}$). In two dimensions, these cells were coupled such that $V$ evolved spatiotemporally, obeying the following reaction-diffusion equation:

$$\frac{\partial V}{\partial t} = \nabla \cdot (D \nabla V) - \frac{I_{ion} + I_{stim}}{C_m}, \tag{2}$$

where $D$ is the diffusion tensor. In simple two-dimensional (2D) monolayer systems, in the absence of anisotropy, the diffusion tensor takes on a diagonal form with identical elements. Thus, $D$ reduced to a scalar in our calculations, with value $0.00012 \ \mathrm{cm^2/ms}$. This resulted in a signal conduction velocity of $22.2 \pm 3.4 \ \mathrm{cm/s}$. Furthermore, in 2D, $I_{K,ACh}$ was considered to be constitutively active, in consonance with the results from in vitro experiments (*Majumder et al., 2016*).

The optogenetic tool used in the numerical studies was a model of *Chlamydomonas reinhardtii* channelrhodopsin-2 mutant H134R, adopted from the studies of *Boyle et al. (2013)* The parameter set used in our studies was exactly the same as that reported by *Boyle et al. (2013)*, with irradiation intensity $0.30 \ \mathrm{mW/mm^2}$ to qualitatively mimic the large photocurrent produced upon illuminating a neonatal rat atrial cardiomyocyte monolayer.

In order to be consistent with the in vitro experiments, we prepared 10 different simulation domains, composed of neonatal rat atrial cardiomyocytes with $17\%$ randomly distributed cardiac fibroblasts (*MacCannell et al., 2007*) and with natural cellular heterogeneity, modelled as per (*Majumder et al., 2016*). Our analysis was performed by taking into consideration the results obtained from all 10 in silico monolayers. In order to demonstrate phenomena such as manipulation of spiral core size and position, we designed several 'illumination' patterns, in the form of a sequence of filled circles of desired radii and projected these patterns on to each of our in silico monolayers. Depending on the aim of the study, we adjusted the durations for which the light was 'on' (i.e. light interval), or 'off' (i.e. dark interval).

## Preparation of CatCh-expressing monolayers

All animal experiments were reviewed and approved by the Animal Experiments Committee of the Leiden University Medical Center and conformed to the Guide for the Care and Use of Laboratory Animals as stated by the US National Institutes of Health. Monolayers of neonatal rat atrial cardiomyocytes expressing $Ca^{2+}$-translocating channelrhodopsin (CatCh) were established as previously described (*Feola et al., 2017*). Briefly, the hearts were excised from anesthetized 2-day-old Wistar rats. The atria were cut into small pieces and dissociated in a solution containing collagenase type I (Worthington, Lakewood, NJ) and 18,75 Kunitz/ml DNase I (Sigma-Aldrich, St. Louis, MO). The resulting cell suspension was enriched for cardiomyocytes by preplating for 120 min in a humidified incubator at $37°C$ and 5% $CO_2$ using Primaria culture dishes (Becton Dickinson, Breda, the Netherlands). Finally, the cells were seeded on round glass coverslips ($d = 15\mathrm{mm}$; Gerhard Menzel, Braunschweig, Germany) coated with fibronectin (100 $\mu$g/ml; Sigma-Aldrich) to establish confluent monolayers. After incubation overnight in an atmosphere of humidified 95% air- 5% $CO_2$ at $37°C$, these monolayers were treated with Mitomycin-C (10 µg/ml; Sigma-Aldrich) for 2 hr to minimize proliferation of the non-cardiomyocytes. At day 4 of culture, the neonatal rat atrial cardiomyocyte monolayers were incubated for 20 – 24 hr with CatCh-encoding lentiviral particles at a dose resulting in homogeneous transduction of nearly 100% of the cells. Next, the cultures were washed once with phosphate-buffer saline, given fresh culture medium and kept under culture conditions for 3 – 4 additional days.

## Optical mapping and optogenetic manipulation

Optical voltage mapping was used to investigate optogenetic manipulation of spiral wave dynamics in the CatCh-expressing monolayers on day 7 of culture by using the voltage-sensitive dye di-4-ANBDQBS (52.5 $\mu$M final concentration; ITK diagnostics, Uithoorn, the Netherlands), as described previously (*Feola et al., 2017*). In summary, optical data were acquired using a MiCAM ULTIMA-L imaging system (SciMedia, Costa Mesa, CA) and analyzed with BrainVision Analyzer 1101 software (Brain vision, Tokyo, Japan). Only monolayers characterized by uniform AP propagation at 1 Hz pacing and homogeneous transgene expression were included for the following optogenetic investigation. CatCh was locally activated by using a patterned illumination device (Polygon400; Mightex Systems, Toronto, ON) connected to a 470-nm, high-power collimator light-emitting diode (LED)

source (type-H, also from Mightex Systems). PolyLite software (Mightex Systems) was used to control the location and movement of the areas of illumination. Before local optogenetic manipulation, reentry was induced by light-based stimulations (n = 18). Reentrant waves that were stable for >1 s were exposed to a circular light spot ($d = 3\mathrm{mm}$) at different targeted locations within the monolayer for $200 - 250 \ \mathrm{ms}$ at $0.3 \ \mathrm{mW/\ mm^2}$.

### Image processing for the videos in vitro A

Optical voltage signal was processed with spatial and temporal derivative filters to allow visualization of electrical wave propagation during the application of the light spots.

### Image processing for the videos in vitro B

The data were sampled according to the location of the applied light spot. Next, for each sample set, we obtained the lowest background intensity recorded by the individual pixels over time and constructed a two-dimensional array with these values as elements. Next, we subtracted this array from each frame in the sample set and reassembled the processed datasets to make a complete video.

## Acknowledgements

This study was supported by The Netherlands Organization for Scientific Research (Vidi grant 91714336) and the European Research Council (Starting grant 716509), both to DAP. Additional support was provided by Ammodo (to DAP and AAFdV). RM would like to thank Dr. Patrick Boyle for discussions. IF would like to thank Cindy Bart and Annemarie Kip for their assistance with the animal experiments and lentiviral particles production.

## Additional information

### Funding

| Funder | Grant reference number | Author |
| --- | --- | --- |
| ZonMw | VIDI 917143 | Daniel A Pijnappels |
| H2020 European Research Council | Starting grant 716509 | Daniel A Pijnappels |
| Ammodo | 2012-Pijnappels | Antoine AF de Vries Daniel A Pijnappels |

The funders had no role in study design, data collection and interpretation, or the decision to submit the work for publication.

### Author contributions

Rupamanjari Majumder, Conceptualization, Data curation, Software, Formal analysis, Supervision, Validation, Investigation, Visualization, Methodology, Writing—original draft, Project administration, Writing—review and editing; Iolanda Feola, Conceptualization, Data curation, Formal analysis, Validation, Investigation, Visualization, Methodology, Writing—original draft, Project administration, Writing—review and editing; Alexander S Teplenin, Investigation, Visualization, Methodology, Writing—original draft, Writing—review and editing; Antoine AF de Vries, Resources, Funding acquisition, Methodology, Writing—original draft, Writing—review and editing; Alexander V Panfilov, Conceptualization, Supervision, Funding acquisition, Writing—original draft, Project administration, Writing—review and editing; Daniel A Pijnappels, Conceptualization, Resources, Supervision, Funding acquisition, Writing—original draft, Writing—review and editing

### Author ORCIDs

Rupamanjari Majumder (iD) http://orcid.org/0000-0002-3851-9225
Iolanda Feola (iD) http://orcid.org/0000-0001-9645-6384

Alexander S Teplenin (iD) http://orcid.org/0000-0001-7841-376X
Daniel A Pijnappels (iD) http://orcid.org/0000-0001-6731-4125

## Ethics

Animal experimentation: All animal experiments were reviewed and approved by the Animal Experiments Committee of the Leiden University Medical Center (AVD 116002017818) and performed in accordance with the recommendations for animal experiments issued by the European Commission directive 2010/63.

## Decision letter and Author response

Decision letter https://doi.org/10.7554/eLife.41076.017
Author response https://doi.org/10.7554/eLife.41076.018

## Additional files

### Supplementary files

• Source data 1.
DOI: https://doi.org/10.7554/eLife.41076.014

• Transparent reporting form
DOI: https://doi.org/10.7554/eLife.41076.015

All data and code generated or analysed during this study are included in the manuscript and supporting files.

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
