## [Decision Letter]

[Editors’ note: this article was originally rejected, but the authors appealed and revised for further consideration.]

Thank you for submitting your work entitled "Optogenetics enables real-time spatiotemporal control over spiral wave dynamics in an excitable cardiac system" for consideration by *eLife*. Your article has been reviewed by two peer reviewers, and the evaluation has been overseen by a Reviewing Editor and a Senior Editor. The following individuals involved in review of your submission have agreed to reveal their identity: Gil Bub (Reviewer #2).

Our decision has been reached after consultation between the reviewers, the Reviewing Editor, and the Senior Editor. Based on these discussions and the individual reviews below, we regret to inform you that your work will not be considered further for publication in *eLife*.

The work shows the ability to move spirals waves of cardiac activity in tissue culture using optogenetic methods to hyperpolarize and modify excitability in localized regions of the culture. The optogenetic techniques were used by the same group in earlier papers to terminate spiral waves. However, this specific technique to move the core location of spiral wave rotation in cardiac tissue had not been used previously. Motion of the spiral to the boundary of a region is a novel finding. The experimental work is accompanied by simulations that show similar behaviors to the experiments. These results do appear to be valid, but they could have been expected based on earlier work in this field by this group and others.

The paper also did not provide adequate citation to other work dealing with control of spirals in excitable medium, and did not develop ways in which the work could be implemented in vivo to help control arrhythmia in people.

After a good deal of back and forth, and consultation with an *eLife* Senior Editor concerning *eLife* editorial policies, the consensus opinion is that this is not sufficiently novel to merit publication in *eLife*.

*Reviewer #1:*

The authors use patterned optogenetic stimulation to drag spiral wave cores in cultured monolayers of cardiomyocytes. They call the protocol the "Attract-Anchor-Drag" (AAD) technique. The authors combine experiments with extensive numerical simulations of a conductance-based cardiac model. They explore the conditions under which the spiral core follows the optical stimulus, by varying (in silico) the radius, duty cycle, and timing of the illumination pulses relative to the rotor circulation. They show that spiral waves can be suppressed either by mutual annihilation of two counter-rotating waves or by dragging a wave to an inexcitable boundary.

Technical merits:

From a technical perspective, this work is a modest variation on several other published works on optogenetic control of excitable cells. The authors cite their own prior work (Feola et al., 2017), but not with sufficient prominence to make clear that the entire technical setup has already been published. The authors neglect to cite several other closely related works:

From the authors' group:

Bingen et al., 2014.

From other groups:

Burton et al., 2015.

Entcheva and Bub, 2016.

The authors cite a work by Zhang et al. on optical electrophysiology in HEK cells, but that paper did not have any spatial control. A more appropriate citation would be:

McNamara et al., 2016.

Scientific merits:

The scientific justification for the study is the experimental demonstration that spiral waves can be manipulated by dragging an inexcitable core. The authors portray this as an important discovery, but in truth the degree of surprise here is limited: given that a spiral wave can anchor at an inexcitable defect, one should not be surprised that gradual motion of the defect will drag the spiral wave.

While the authors are to be commended for exploring (in silico) the impact of the size, timing, and motion parameters of the spot on the spiral wave motion (Figure 3), all results are reported in physical units, giving little insight into how these parameters might change in a different system, e.g. real cardiac tissue where conduction velocity and gap junction coupling are very different from in vitro. It is not clear what generalizable insights can be gained from these results.

The other experimental results demonstrate that spiral waves can be annihilated by dragging to the boundary of a dish (Figure 2), or by coalescing two counter-propagating waves (Figure 4). These results are interesting, but also not particularly surprising and of uncertain relevance to treatment of spiral wave defects in vivo.

Overall, the experimental work and analysis seems thin and anecdotal, comprising video stills from three videos (Figures 1, 2, 4). It is disappointing that the simulations in Figure 5 are not accompanied by experiments.

The broader scientific justification is a general statement that these techniques "could have significant meaning in terms of a better mechanistic understanding of cardiac arrhythmias and improvement of existing and development of new treatment modalities." (stated again at the end of the Introduction and Discussion). While it is generally true that optical control of cardiac dynamics is an interesting topic, these statements seem to be insufficiently justified by the present work. It is not clear how one would apply the learning from the present experiments to work in vivo or to improved therapies.

Previous work from the senior authors showed that widefield optogenetic stimulation could eliminate spiral wave patterns (Bingen et al., 2014). If the authors want to emphasize the utility of their new approach for spiral wave abolition, they should discuss the relative advantages or limitations of these two strategies. Moreover, many examples of AAD spiral wave elimination described here rely on colliding the spiral core with a boundary. It is not obvious how such a strategy would be implemented in vivo.

The simulation work in Figure 4 (Video 4) is beginning to approach interesting questions, e.g. under what circumstances does combining two spirals lead to multiarm spirals vs. elimination? If brought together and released, how do rotors interact with each other? How does this interaction depend on phase offset, direction of circulation, and number of arms? A more thorough experimental exploration of these questions would considerably improve the overall scholarship and impact of the work.

*Reviewer #2:*

The article by Majumder et al. demonstrate a novel control method for steering spiral waves in a biological excitable medium. The experiments and theory are convincing.

My only major concern is that the authors may have misstated the impact of previous studies. The authors state in their final paragraph:

"Previous studies also demonstrated alternative methods to 'control' spiral-wave dynamics in excitable media, for example,… However, the principal limitation of these methods is that they are indirect, giving reasonable control over the initial and final states of the system, with little or no control in between."

I don't agree with this statement. First, the submitted work does not show a higher degree of control than other published studies as the authors apply a perturbation and observe the results as opposed to using closed loop feedback control. The authors may mean that previous control systems rely on delivering timed pulses which fall in defined phases of the spiral wave (but I don't see why this indicates a lack of control of the system). Also, there are un-referenced published examples of feedback control of spiral wave motion (e.g. Schlesner et al., 2008; Sakurai et al., 2002) that show very precise spiral wave control. In addition, the general idea of anchoring a spiral with light has been demonstrated in an experimental system which would require relatively small modification to show spiral wave steering (Steinbock and Muller, 1993).

The figures should be clarified. For example, in Figure 2, panels are shown without indicating the precise timing (i.e. B1, B2, B3 etc.), which would help readers understand the observed dynamics. Also, exactly how the spiral translates linearly is unclear from the figure, as the speed of the moving spot is unknown. It would be helpful to note the speed of the spot relative to the speed of wave propagation.

---

## [Author Response]

We now address the concerns around novelty by including in vitro data to match the simulations of Figure 5. We have also rewritten significant portions of the Introduction and the Discussion section to better highlight the importance and novelty of our study. In order to address the comment on the in vivo translatability of our Attract-Anchor-Drag (AAD) method, we have added a special sub-section called ‘Clinical translation’, to the Discussion. Briefly, in vivo translation of AAD would require the following steps: (1) expression of the light-sensitive depolarizing ion channel in the living heart and (2) technology to illuminate the living heart.

Fiber optics allows one to bring light from an external source to inside the heart. A small local light source can also be brought to the heart via a catheter. Work on it is under way. However, this is mainly a technical issue and can easily be realized with biomedical engineering in the near future. We do not include it in this manuscript because that would not only require the development, testing and implementation of a completely new experimental setup (as proposed in the Discussion section of the paper), but would also need us to explore the method without additional research in order to optimally prepare for an animal study. We, therefore, feel that such in vivo translation is outside the scope of our current study and would be somewhat untimely. They require a dedicated future study in order to provide additional insight in a constructive, efficient and responsible manner. Our paper actually provides the basis for the design and execution of studies in the whole heart, now proof-of-principle, with identification of the key parameters. Lastly, there is also a scientific concern: If the AAD method is realized, why would it be better than the current arrhythmia termination methods? We extensively discuss this issue in the ‘Clinical translation’ section.

The work shows the ability to move spirals waves of cardiac activity in tissue culture using optogenetic methods to hyperpolarize and modify excitability in localized regions of the culture.

We use optogenetic methods to depolarize, rather than to hyperpolarize, localized regions of the culture. This is a very important distinction, given that depolarization and hyperpolarization produce different biophysical effects on cardiac tissue and lead to contrary spatiotemporal behaviour of electrical spiral waves.

The optogenetic techniques were used by the same group in earlier papers to terminate spiral waves. However, this specific technique to move the core location of spiral wave rotation in cardiac tissue had not been used previously. Motion of the spiral to the boundary of a region is a novel finding. The experimental work is accompanied by simulations that show similar behaviors to the experiments.

Indeed, we made use of previously established optogenetic techniques to demonstrate a new approach, called Attract-Anchor-Drag" (AAD) control, to provide novel general insight into spiral wave dynamics. We consider these findings important as they came from actual experiments in a complex biological system (i.e. cardiac tissue) and the mechanisms were studied in detail in corresponding computer simulations, which indicates that AADcontrol of spiral wave is indeed a very robust approach. From a general point of view, a spiral wave is a nonlinear dynamical object whose motion is defined by the parameters of the system. To manipulate its dynamics, one needs to manipulate the system. In our manuscript, however, we show, for the first time, not only the possibility to control the precise movement of a spiral wave without applying a global external perturbation to the system, but also how such control can be gained and exploited.

These results do appear to be valid, but they could have been expected based on earlier work in this field by this group and others.The paper also did not provide adequate citation to other work dealing with control of spirals in excitable medium, and did not develop ways in which the work could be implemented in vivo to help control arrhythmia in people.

We agree that our results appear straightforward and perhaps could have been expected, based on generic spiral wave theory. However, to the best of our knowledge, no one has yet published data on guided dragging of spiral waves in spatially extended, complex biological systems, such as we do in cardiac tissue. The phenomenon closest to what we show, was reported by Steinbock et al. in a Belousov–Zhabotinsky (BZ) reaction, where they anchored a scroll wave to a thin glass rod and dragged it by moving the rod. However, those results cannot be directly extended to our observations because properties of chemical systems such as the BZ reaction essentially differ from those of cardiac tissue due to the absence of spatial symmetries in cardiac tissue caused by its inherently irregular discrete cellular structure, the lack of inhibitor diffusion in cardiac tissue and different spatial scales of the systems. It is also not obvious that discrete application of light spots outside the core of a spiral wave should only end up in an anchored state of the spiral wave, since the probability of attraction is determined by the initial conditions as well as by the strength of successive perturbations. Thus, definitive proof that it is possible to control spiral wave dynamics in cardiac tissue by targeting the spiral wave core does require experimental confirmation. Hence the results are not so obvious.

Based on the suggestion of the Editor and the reviewers, we have tried to improve the presentation of the work of other groups. In particular, we now re-write the Introduction section, with new references as follows:

“Self-organization of macroscopic structures through atomic, molecular or cellular interactions is characteristic of many non-equilibrium systems.[…] In particular, in the heart, tight control of spiral waves may allow restoration of normal wave propagation.”

Regarding implementation of our method in vivo to help control arrhythmia in patients, please see our fourth response to reviewer #1.

After a good deal of back and forth, and consultation with an eLife Senior Editor concerning eLife editorial policies, the consensus opinion is that this is not sufficiently novel to merit publication in eLife.

We thank the editorial team of *eLife* for granting us an opportunity to resubmit our manuscript, based on modifications suggested by the Editors and reviewers. We kindly ask the *eLife* review team to reconsider their decision based on this rebuttal and the revised version of our manuscript.

Reviewer #1:[…] From a technical perspective, this work is a modest variation on several other published works on optogenetic control of excitable cells. The authors cite their own prior work (Feola et al., 2017), but not with sufficient prominence to make clear that the entire technical setup has already been published. The authors neglect to cite several other closely related works:From the authors' group:Bingen et al., 2014.From other groups:Burton et al., 2015.Entcheva and Bub, 2016.The authors cite a work by Zhang et al. on optical electrophysiology in HEK cells, but that paper did not have any spatial control. A more appropriate citation would be: McNamara et al., 2016.

We agree that our study relies on previously published optogenetics methodology. The current paper is, however, not a methodological one. Instead, we exploit the unique possibilities offered by optogenetics to tackle a classical and practically important problem of how to gain control of spiral wave location. We propose a new way to achieve this control and demonstrate it in vitro (i.e. in cultured cardiac tissue) on the basis of in silico parameter predictions. To the best of our knowledge this is the first successful solution to this problem.

In our view the mechanisms involved are general for any system that is capable of supporting spiral waves. What we have demonstrated here, is the fundamental dynamical interaction between spiral waves and movable temporal defects, with consequential, real-time, and precise spatiotemporal control over spiral wave dynamics in spatially-extended biological systems. Optogenetics merely provides one possible means to create such movable defects. One may use other methods to achieve the same goal in other biological/physical/chemical systems. Thus, we think that this is a generic finding suiting the broad readership of *eLife*.

We have now re-written the Introduction section to emphasize the more general nature of our findings. Also, we have updated our reference list to include more relevant literature, including the ones suggested by the referee.

Scientific merits:The scientific justification for the study is the experimental demonstration that spiral waves can be manipulated by dragging an inexcitable core. The authors portray this as an important discovery, but in truth the degree of surprise here is limited: given that a spiral wave can anchor at an inexcitable defect, one should not be surprised that gradual motion of the defect will drag the spiral wave.

We agree that the effect reported in our study is based on the fundamental properties of spiral wave dynamics: attraction and anchoring to inexcitable obstacles. What we find exciting is that this simple straightforward methodology works not only in silico, but in real experiments using a complex biological system, such as cardiac tissue, thereby proving the robustness of the effect. In our view, the fact that the effect is a result of very basic mechanisms is not a downside to our study at all. In fact, this should be considered as a strong point, as it indicates that the effect has fundamental roots and may be realized in a wide range of conditions and systems. Furthermore, regarding the alleged limited degree of surprise we would like the referee to note the following:

We agree that our results appear straightforward and perhaps could have been expected, based on generic spiral wave theory. However, to the best of our knowledge, no one has managed to drag spiral waves in spatially extended, complex biological systems, such as we do in cardiac tissue. The phenomenon closest to what we show, was reported by Steinbock et al. in a Belousov–Zhabotinsky (BZ) reaction, where they anchored a scroll wave to a thin glass rod and dragged it by moving the rod. However, those results cannot be extended to our observations as properties of chemical systems such as the BZ reaction essentially differ from those of cardiac tissue because of the absence of spatial symmetries in cardiac tissue caused by its inherently irregular discrete cellular structure, the lack of the inhibitor diffusion in cardiac tissue and different spatial scales of the systems. We would like emphasize that application of light-induced defects was done in a discrete manner, not in the gradual fashion inferred by the referee. It is not obvious that discrete application of light spots outside the core of a spiral wave should only end up in an anchored state, since the possibility of attraction is determined by the initial conditions and by the strength of successive perturbations. Thus, by showing, in cultured cardiac tissue, that spiral wave dynamics can be controlled via targeting of the spiral wave core, we have made an important contribution to the understanding of spiral wave behavior in complex systems.

We have now included the following paragraph to the Discussions section:

“Although apparently straightforward, and, to some extent, predictable on the basis of generic spiral wave theory, our results are not at all obvious. […] By showing, in cultured cardiac tissue, that spiral wave dynamics can be controlled by targeting the spiral wave core, we have been able to experimentally confirm pre-existing ideas about the behavior of spiral waves in complex systems.”

While the authors are to be commended for exploring (in silico) the impact of the size, timing, and motion parameters of the spot on the spiral wave motion (Figure 3), all results are reported in physical units, giving little insight into how these parameters might change in a different system, e.g. real cardiac tissue where conduction velocity and gap junction coupling are very different from in vitro. It is not clear what generalizable insights can be gained from these results.

We thank the referee for his/her commendation. Our in silico studies were carried out to guide the in vitro experiments, which required real numbers in real units. Therefore, the numbers that we have presented in the paper are specific for a monolayer of optogenetically modified neonatal rat atrial cardiomyocytes. We have also performed several simulations in the Ten Tusscher-Panfilov 2006 model for human ventricular tissue, which gave essentially the same effects. However, to prevent losing focus, we decided to restrict ourselves in both the in vitro and the in silico studies to neonatal rat atrial tissue and hence did not include the human ventricular data in the manuscript. Here, we demonstrate not only the possibility to control the precise movement of a spiral wave without globally affecting the system, but also how such control can be gained and applied, and that the message is generalizable.

The other experimental results demonstrate that spiral waves can be annihilated by dragging to the boundary of a dish (Figure 2), or by coalescing two counter-propagating waves (Figure 4). These results are interesting, but also not particularly surprising and of uncertain relevance to treatment of spiral wave defects in vivo.

We agree that the illustrations of Figure 2 and Figure 4 are logical and straightforward, based on our observation that a spiral can be dragged by a moving obstacle. However, we decided to show these results in order to provide examples of different possibilities for removing spiral waves by manipulating the core.

Clinical application of optogenetics in the heart remains challenging and several groups including ours are working on how to make it more feasible. Regarding the dragging phenomenon that we report in our paper, we see the following possibilities: during a clinical procedure the cardiologist needs to precisely position the ablation catheter in order to terminate a reentrant arrhythmia. Herein lies the advantage of our method, which, as demonstrated in our paper, relies on the spiral wave to detect the illuminated spot by itself. It is not too difficult to determine the approximate location of a reentrant circuit during an arrhythmia. Illuminating a region in the heart that is reasonably close to the scroll wave will cause the scroll wave to anchor around it. The anchored scroll wave can then be driven to extinction by subsequent dragging. Creation of such illuminated spot is practically possible via a catheter. Compared to constant-in-time global or local application of light in classical implementations of cardiac optogenetics, dragging may have some advantages. In human ventricles, for example, 470 nm light cannot penetrate the full transmural thickness of cardiac tissue, thus limiting the possibilities for optogenetic defibrillation. However, dragging the scroll wave by anchoring one end of the scroll filament to the applied light spot can still be a good strategy for arrhythmia termination via optogenetic means. Also, if a scroll wave is detected in a part of the heart muscle, which is difficult to reach by an ablation catheter, then one may use distant light source to guide the spiral wave out to a more open area where it can be subsequently terminated.

In order to give the readers more insight into how spiral wave dragging can be employed in the treatment of arrhythmias in vivo, we have now included the following new sub-section called ‘Clinical translation’ to the Discussion section:

“Since we demonstrate AAD control method in a cardiac tissue system, a logical question would be, how to envision the application of this principle to the real heart in order to treat arrhythmias? Currently this topic faces major challenges. […] For example, in younger patients that are expected to undergo periodic repetitive ablation for termination of reoccurring arrhythmias of unknown origin, the non-destructive nature of AAD may prove to be more desirable than the cumulative widespread destruction of cardiac tissue by radiofrequency or cryoballoon ablation.”

Overall, the experimental work and analysis seems thin and anecdotal, comprising video stills from three videos (Figures 1, 2, 4). It is disappointing that the simulations in Figure 5 are not accompanied by experiments.

We have tried our best to supply representative figures that illustrate the phenomenon in question. Still, the intrinsic limitations of video stills make it hard to fully capture dynamic processes. It is for this reason that we provided annotated videos of the results presented in Figures 1, 2, 4, and 5 of the original manuscript. Regarding the simulations of Figure 5, we did not initially provide experimental validation of these results because the mechanism of this phenomenon is the same as that for the experiments of Figure 4. Based on the previous example, we hoped to have convinced the reader that our simulations do make valid predictions and decided here to use the opportunity offered by predictive in silico modeling instead of performing the more challenging experimental studies. However, we now include a new movie (Video 5) and a representative figure from our in vitro studies to experimentally prove the concept of (old) Figure 5, together with the following paragraph in the Results section:

“A similar strategy applied in vitro enabled successful termination of a complex reentry pattern characterized by 4 phase singularities. […] The single spiral can then be dragged to termination (Figure 5C6-7) as in Figure 2.”

And the following update of the legend of Figure 5:

“AAD control of multiple spiral wave cores in favor of termination of complex reentrant patterns. […] Video 5 shows the same process, in vitro, but for stable reentry with 4 phase singularities.”

The broader scientific justification is a general statement that these techniques "could have significant meaning in terms of a better mechanistic understanding of cardiac arrhythmias and improvement of existing and development of new treatment modalities." (stated again at the end of the Introduction and Discussion). While it is generally true that optical control of cardiac dynamics is an interesting topic, these statements seem to be insufficiently justified by the present work. It is not clear how one would apply the learning from the present experiments to work in vivo or to improved therapies.Previous work from the senior authors showed that widefield optogenetic stimulation could eliminate spiral wave patterns (Bingen et al., 2014). If the authors want to emphasize the utility of their new approach for spiral wave abolition, they should discuss the relative advantages or limitations of these two strategies. Moreover, many examples of AAD spiral wave elimination described here rely on colliding the spiral core with a boundary. It is not obvious how such a strategy would be implemented in vivo.

Please see our fourth response to reviewer #1..

The simulation work in Figure 4 (Video 4) is beginning to approach interesting questions, e.g. under what circumstances does combining two spirals lead to multiarm spirals vs. elimination? If brought together and released, how do rotors interact with each other? How does this interaction depend on phase offset, direction of circulation, and number of arms? A more thorough experimental exploration of these questions would considerably improve the overall scholarship and impact of the work.

In general, annihilation of spirals by collision of the cores requires the spirals to have opposite topological charge, whereas spirals with the same topological charge (i.e. rotating in the same direction) do not annihilate. We agree that using our methodology one can study various regimes of spiral wave interactions. We have demonstrated that, by manipulating the rotational phase of one of the two spirals, one can either result in the formation of a multi-armed spiral or lead to spiral wave elimination. Here, we could have provided a detailed study of the parameters determining the outcome of the interaction of two spiral waves. However, such parameter surveys would cause us to deviate from the main research question here, i.e., how to control motion of spiral waves by manipulating the core. Also, such studies should better be performed in silico, owing to the fact that parameters can be more easily and precisely adjusted in silico than in vitro.

Reviewer #2:The article by Majumder et al. demonstrate a novel control method for steering spiral waves in a biological excitable medium. The experiments and theory are convincing.My only major concern is that the authors may have misstated the impact of previous studies. The authors state in their final paragraph:"Previous studies also demonstrated alternative methods to 'control' spiral-wave dynamics in excitable media, for example,… However, the principal limitation of these methods is that they are indirect, giving reasonable control over the initial and final states of the system, with little or no control in between."I don't agree with this statement. First, the submitted work does not show a higher degree of control than other published studies as the authors apply a perturbation and observe the results as opposed to using closed loop feedback control. The authors may mean that previous control systems rely on delivering timed pulses which fall in defined phases of the spiral wave (but I don't see why this indicates a lack of control of the system). Also, there are un-referenced published examples of feedback control of spiral wave motion (e.g. Schlesner et al., 2008; Sakurai et al., 2002) that show very precise spiral wave control. In addition, the general idea of anchoring a spiral with light has been demonstrated in an experimental system which would require relatively small modification to show spiral wave steering (Steinbock and Muller, 1993).

We thank the referee for pointing out some omissions in the coverage of earlier work. We have now carefully re-written the Introduction section to undo these omissions (see paragraph below) and included all necessary references to the revised version of our manuscript. We agree that the work of Steinbock and Muller is indeed a precursor to our results, but in a different, non-biological setting. Please note that results from Belousov–Zhabotinsky (BZ) reactions cannot be directly translated to cardiac tissue due to lack of inhibitor diffusion in cardiac tissue. Moreover, in the photosensitive BZ reaction, wave propagation is inhibited by light, whereas, in our system it can not only initiate new waves, but also inhibit their propagation due to complex dynamics of the cardiac sodium current. Accordingly, the findings of our study come with substantially new insight compared to the previous work.

New Introduction:

“Self-organization of macroscopic structures through atomic, molecular or cellular interactions is characteristic of many non-equilibrium systems. […] In particular, in the heart, tight control of spiral waves may allow restoration of normal wave propagation.”

We would like to thank the referee for his suggestion to consider the work of Schlesner et al., 2008, as a reference for published articles on feedback control over spiral wave motion because it seems to be the only article demonstrating motion of a spiral wave along an arbitrary path. Although highly interesting, the method of Schlesner and co-workers unfortunately is too difficult to realize in practice in biological substrates. It requires application of a *global* weak stimulus when the front of the spiral wave is *tangential* to this predefined trajectory. The tangentiality condition is difficult to ascertain in practice due to large amount of noise and low spatial resolution in cardiac tissue imaging.

We have also added the following text to the Discussion:

“While it is an established theory that the properties of the core of a spiral wave determine its overall dynamics (Beaumont et al., 1998; Krinsky, 1978), the corollary that manipulation of spiral wave cores should lead to full spatiotemporal control over wave dynamics in excitable media, appears to be unexpectedly non-obvious and understudied (see below). […] However, (Burton et al., 2015) use dynamic control to modulate spiral wave chirality, which is markedly different from what we study and prove, namely spiral wave dragging.”

And:

“In the past, researchers also employed different kinds of spatiotemporal feedback interactions to control spiral wave dynamics, albeit in non-biological media (Sakurai et al., 2002; Schlesner et al., 2008). […] However, (Ke et al., 2015) demonstrate a process that is continuous in time, as opposed to the discrete nature of our method, which makes it more flexible.”

And:

“Although apparently straightforward, and, to some extent, predictable on the basis of generic spiral wave theory, our results are not at all obvious. […]By showing, in cultured cardiac tissue, that spiral wave dynamics can be controlled by targeting the spiral wave core, we have been able to experimentally confirm pre-existing ideas about the behavior of spiral waves in complex systems.”

The figures should be clarified. For example, in Figure 2, panels are shown without indicating the precise timing (i.e. B1, B2, B3 etc.), which would help readers understand the observed dynamics. Also, exactly how the spiral translates linearly is unclear from the figure, as the speed of the moving spot is unknown. It would be helpful to note the speed of the spot relative to the speed of wave propagation.

We thank the referee for this suggestion. We have now updated the legend of Figure 2 as follows:

“AAD control of a spiral wave core in favor of termination. […] A video demonstrating the complete process of dragging a spiral wave core from the center of the monolayer to the left border, causing its termination, is presented in Video 2. (Time t=0 ms denotes the moment when the light is applied.)”

We agree that the linear translation of the spiral may be unclear from our representation of Figure 2. That is why we have provided the video. Furthermore, we have added timestamps to each of the figures following the reviewer’s suggestion, and have included the numbers for τ light, τ dark and period of the reentrant activity in the legend of Figure 2, which give an estimate of the speed of the spot relative to the speed of wave propagation.